# Learn and Consolidate: Continual Adaptation for Zero-Shot and Multilingual Neural Machine Translation

**Kaiyu Huang**[1], **Peng Li**[*,1,3], **Junpeng Liu**[4], **Maosong Sun**[2], **Yang Liu**[*1,2,3]

[1]Institute for AI Industry Research (AIR), Tsinghua University, Beijing, China
[2]Dept. of Comp. Sci. & Tech., Institute for AI, Tsinghua University, Beijing, China
[3]Shanghai Artificial Intelligence Laboratory, Shanghai, China
[4]Dalian University of Technology, Liaoning, China
{huangkaiyu,lipeng}@air.tsinghua.edu.cn; liuyang2011@tsinghua.edu.cn

## Abstract

Although existing multilingual neural machine translation (MNMT) models have demonstrated remarkable performance to handle multiple translation directions in a single model and achieved zero-shot translation between language pairs unseen in training, they still suffer from relatively poor translation qualities for some language pairs. A practical scenario is that how to continually update MNMT models for both supervised and zero-shot translations when limited new data arrives. To this end, we propose a two-stage approach that encourages original models to acquire language-agnostic multilingual representations from new data, and preserves the model architecture without introducing parameters. Experimental results and further analysis demonstrate that our method can efficiently improve performance of existing MNMT models in translation directions where they are initially weak, and mitigates the degeneration in the original well-performing translation directions, offering flexibility in the real-world scenario.[1]

## 1 Introduction

Existing multilingual neural machine translation (MNMT) models, such as mBART (Liu et al., 2020) and M2M-100 (Fan et al., 2021), have showcased significant advancements to handle multiple translation directions in a single model, especially enabling zero-shot translation directions between languages not encountered during training with implicit cross-lingual transfer (Johnson et al., 2017; Tan et al., 2019; Zhang et al., 2020; Cheng et al., 2022). However, they still perform poorly on language pairs without sufficient parallel corpus (Goyal et al., 2022).

Fortunately, new parallel sentence pairs will continually emerge between high-resource lan-

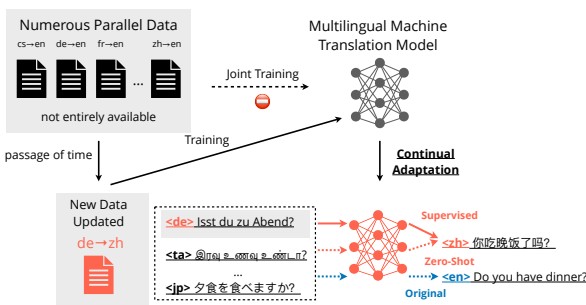

Figure 1: We aim at adapting existing MNMT models to both supervised and zero-shot translation directions with newly available parallel data while maintaining the stronger performance in English-Centric directions, where dashed lines represent no parallel corpus.

guages, e.g., German↔Chinese, which can be used to facilitate translation directions with poor performance through supervised learning (Costa-jussà et al., 2022). In addition to updated data, obtaining parallel data between some languages, e.g., Tamil↔Chinese, encounters difficulties in the world (Siddhant et al., 2022). We aspire to achieve many-to-one translation with cross-lingual transferability of MNMT models in a zero-shot manner (Chen et al., 2022), when the "one" target language related data is updated. As shown in Figure 1, we assume three translation directions for continual adaptation: new supervised translations, new zero-shot translations, and original well-performing translations (typically English-Centric). In this scenario, we aim to continually improve performance for both new supervised and zero-shot translations while retaining previously acquired knowledge in the other translation directions using only new data.

In accordance with this scenario, an intuitive solution is to introduce additional parameters for continual adaptation, regarded as parameter-isolation based methods (He et al., 2021; Madotto et al., 2021). The drawback of these methods is that they increase the size of the base models, and as

---

[*]Corresponding authors: Peng Li (lipeng@air.tsinghua.edu.cn) and Yang Liu (liuyang2011@tsinghua.edu.cn)

[1]Our code is released at https://github.com/koukaiu/LCCA

new data continues to accumulate, the total number of parameters may expand indefinitely (Escolano et al., 2021). To preserve the model architecture, another solution is to leverage the continual learning paradigm (Garcia et al., 2021; Huang et al., 2022b; Gu et al., 2022; Huang et al., 2022c). Although these methods propose multi-objective learning to alleviate the issue of catastrophic forgetting, they necessitate full parameter fine-tuning (Tang et al., 2020), resulting in relatively low training efficiency (Ke and Liu, 2022). Besides, a significant pitfall of the above-mentioned methods is that they lack the ability to promote zero-shot translation.

In this work, we propose a two-stage method consisting of a **l**earning stage and a **c**onsolidation stage for **c**ontinual **a**daptation (LCCA), which efficiently adapts MNMT models to diverse translation directions. We first introduce a flexible pluggable module in the penultimate encoder and decoder layer, respectively, as the multilingual adaptation space. And the introduced multilingual space is optimized with contrastive learning to make representation language-agnostic for zero-shot translations, realizing the learning stage. Then we attempt to compress the additional parameters to the same size of base MNMT models, regarded as a consolidation stage. The second stage adopts the information matrix and collaborative distillation to facilitates original components to learn introduced modules in a specific range. Furthermore, aside from the related components, all the parameters of the original model remain fixed, allowing for a parameter-efficient manner to large-scale MNMT models. The two stages are also an isolated process designed to adapt to diverse application requirements in real-world scenarios.

To sum up, our contributions are as follows:

- We propose LCCA, a two-stage approach that encourages to learn new knowledge for continual adaptation, while mitigating performance degeneration without introducing additional parameters.

- The ability of the original MNMT model to translate into a target language is enhanced via acquiring language-agnostic representation, which improves performance for zero-shot translations in continual learning.

- Experimental results demonstrate the efficiency and flexibility of our approach in adapting various powerful and open-source MNMT

models of different sizes to new parallel data.

## 2 Related Work

**Zero-Shot Translation with MNMT Models** MNMT models have demonstrated their ability to facilitate knowledge transfer across languages and enable zero-shot translations between language pairs that are not covered in training data (Lakew et al., 2018; Tan et al., 2019; Zhang et al., 2020; Chen et al., 2022). To further enhance the performance of zero-shot translation, some studies investigate language-agnostic representations (Arivazhagan et al., 2019a; Pham et al., 2019; Liu et al., 2021) and language-specific features (Wang et al., 2019; Yang et al., 2021) for zero-shot translations. For instance, Pan et al. (2021) align cross-lingual representations using additional dictionaries and contrastive learning. This work continues to explore the potential of multilingual representations, but focusing on addressing the continual adaptation for new zero-shot translations within an well-performed representation space while avoid catastrophic forgetting.

**Continual Learning for MNMT** Some previous methods of continual learning attempt to address the issue of catastrophic forgetting when only the new data is accessible (De Lange et al., 2019; Wu et al., 2021), including replay-based methods (Sun et al., 2019; Tang et al., 2020; Garcia et al., 2021), regularization-based methods (Kirkpatrick et al., 2017; Castellucci et al., 2021; Huang et al., 2022a; Zhao et al., 2022), and parameter-efficient transfer (PET) methods (Bapna and Firat, 2019; Zhu et al., 2021; Huang et al., 2023). On the one hand, the first two methods adopt the multi-objective learning (Thompson et al., 2019; Peng et al., 2020) to balance performance between old and new tasks in continual learning through full parameters tuning, which is inefficient and time-consuming.

On the other hand, PET methods introduce additional task-specific parameters and freeze all original parameters to completely retain performance on previous tasks (Chalkidis et al., 2021; Li and Liang, 2021; Huang et al., 2023). However, they introduce extra parameters to adapt to new tasks and lack the ability to improve zero-shot performance, limiting their application in practice (Dabre et al., 2020). Differing from these approaches, our method mitigates the issue of zero-shot translations only using new data, and achieves efficient continual adaptation without introducing additional parameters.

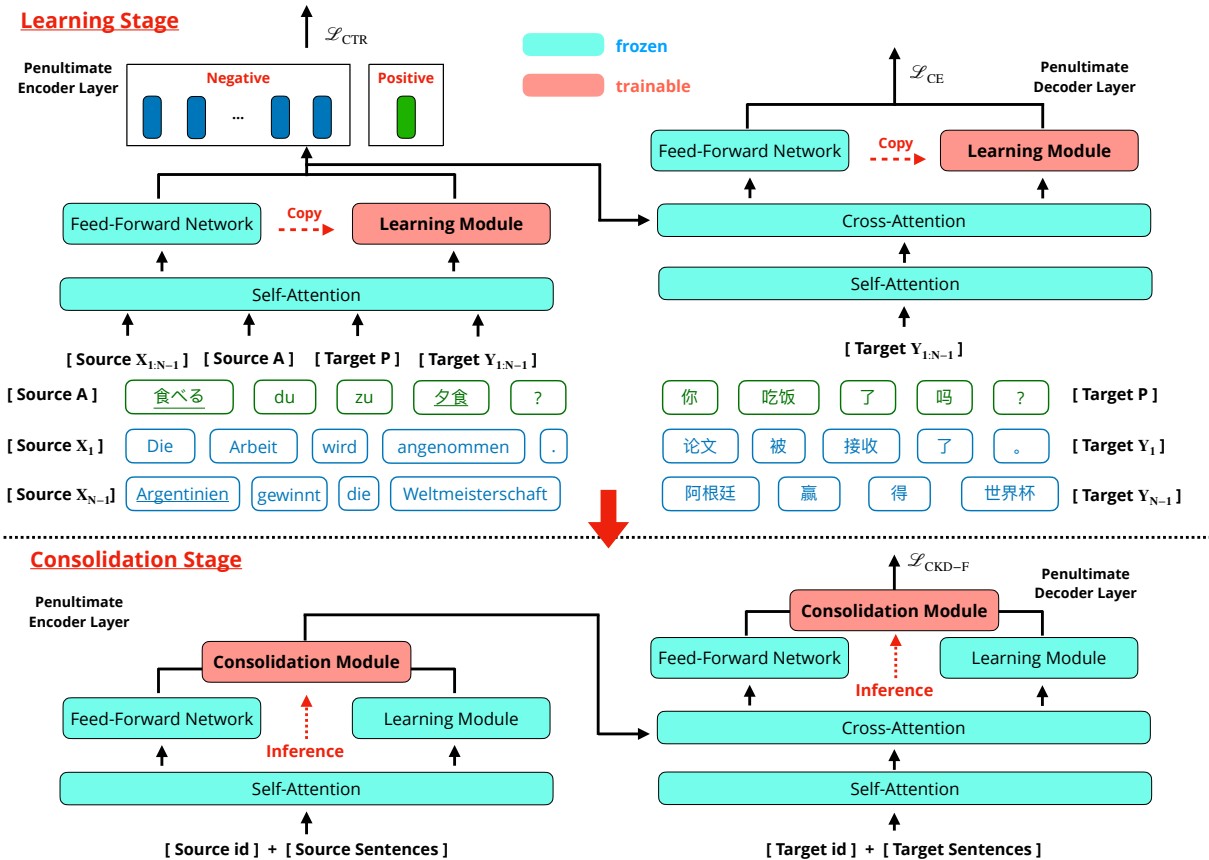

Figure 2: Illustration of our approach LCCA. **[ Source A**, **Target P ]** denotes a positive sample that is the bilingual or code-switching sentence pair, while **Target $Y_{1:N-1}$** denotes $N - 1$ negative samples selected from the same batch for contrastive learning. The blue modules are frozen and the red modules are trainable during their own stage.

This enhanced flexibility makes it well-suited for real-world scenarios.

## 3 Method

Our scenario is to efficiently improve performance for some particular translation directions without compromising previous well-performing translations. To achieve this, we propose a two-stage method to alleviate the issues of continual adaptation in particular translation directions at different stages, as shown in Figure 2. Specifically, we first introduce an additional space to capture language-agnostic features for new target languages, as the learning stage. Then we compress the additional space back into the original models with a collaboration, as the consolidation stage. And we calculate an information matrix to measure the parameters optimization in a smooth region across multiple languages.

### 3.1 Task Definition

Multilingual translation models can provide high-quality translation services on many language pairs and are trained on selecting available parallel data with multiple language pairs. Given the anticipated growth in available data, it becomes possible to continually update the multilingual models for both their original and newly emerging translation directions, as illustrated in Figure 1.

Formally, the training process of an MNMT model commences with an initial set of available parallel data denoted as $D = D_1, ..., D_i, ..., D_M$, encompassing a total of $M$ languages. Each $D_i$ represents the training corpus for the $i$-th language pair. In this framework, the primary MNMT model, given an input sentence $\mathbf{x}$, undergoes optimization by maximizing the log-likelihood $\mathcal{L}$ for the ground-truth sequence $\mathbf{y}$. This is mathematically expressed as:

$$\mathcal{L}_D(\theta) = \sum_{D_i \in D} \sum_{(\mathbf{x}, \mathbf{y}) \in D_i} \log p(\mathbf{y}|\mathbf{x}; \theta) \quad (1)$$

Here, $\theta$ represents the parameters of the MNMT model. For language identification, a specific language token is prepended to the beginning of both source and target sentences, following the convention introduced in prior work (Liu et al., 2020).

We aim to achieve supervised translation directions between high-resource languages ($L_s$ and $L_t$) and zero-shot translation from $M$ languages $L_1, L_2, ..., L_M$ to the target language $L_t$ with the help of only the newly available data $D' = \{\mathbf{x}, \mathbf{y}\}$. Due to the unavailability of the original collection of parallel data, the optimization objective in continual learning is given by:

$$\mathcal{L}_{D'}(\theta) = \sum_{(\mathbf{x}, \mathbf{y}) \in D'} \log p(\mathbf{y}|\mathbf{x}; \theta) \qquad (2)$$

As a result, we aspire to continually update MNMT models for both supervised and zero-shot translations using limited new data.

### 3.2 Learning Stage

One of the crucial steps in this task involves acquiring new knowledge from newly available data. Previous studies show that the FFN layers can be seen as key-value memories and store knowledge in this manner (Sukhbaatar et al., 2019). Thus, the FFN might be a core component which stores cross-lingual transferability for multilingual translations. As shown in Figure 2, to adapt original models to new data, we open up two additional blocks in the hidden layers, as the learning modules. The two blocks are replicated from FFN, thus are the same as corresponding FFN layers before tuning. Considering that the original models normally have a large number of parameters, introducing additional parameters in every layer would increase training costs and reduce training efficiency. Therefore, we only introduce learning modules in the penultimate layer of the original model, as the knowledge stored in the FFN of this layer is closer to the high-level semantic information. To further leverage knowledge from the original model, we combine the original FFN and the learning modules, which can share linguistic knowledge. The fusion output of hidden states $\mathbf{H}_f$ is given by:

$$\mathbf{H}_f = \text{FFN}_{\text{original}}(\mathbf{H}) + \text{FFN}_{\text{learn}}(\mathbf{H}) \quad (3)$$

The learning module is optimized by minimizing a cross-entropy loss of the parallel sequence pairs:

$$\mathcal{L}_{\text{CE}}(\theta) = -\frac{1}{J} \sum_{j=1}^{J} \log p(y_j|\mathbf{y}_{<j}, \mathbf{x}; \theta) \quad (4)$$

where J is the length of the target sentence.

Our scenario also focuses on facilitating the ability of the original MNMT model to translate into a target language, achieving zero-shot translation. We argue that during the learning stage for continual adaptation, the additional learning module in the encoder lacks to align parallel sentences of supervision that can bridge the representation gap across different languages. To this end, we introduce a contrastive learning loss as an auxiliary supervision to learn language-agnostic representations for the encoder. Given a sentence pair $(\mathbf{x}, \mathbf{y})$ from new training data $\mathcal{D}'$, we denote it as a positive example and select a set of target sentences $\{\mathbf{y}_i\}_{i=1}^{N-1}$ from the same batch as negative examples. The contrastive loss is given by:

$$\mathcal{L}_{\text{CTR}} = - \sum_{(\mathbf{x}, \mathbf{y}) \in \mathcal{D}'} \log \frac{e^{\text{sim}(\mathbf{H}_f(\mathbf{x}), \mathbf{H}_f(\mathbf{y}))/\tau}}{\sum_{\mathbf{y}_i} e^{\text{sim}(\mathbf{H}_f(\mathbf{x}), \mathbf{H}_f(\mathbf{y}_i))/\tau}}$$

$$(5)$$

where sim is the cosine similarity function and $\tau$ is the temperature parameter which is set to 0.1.

### 3.3 Consolidation Stage

Due to introduced parameters in the learning stage, the architecture of original models is modified. Not only the model increases the total number of parameters, but also it requires to determine the specific task to which the input sentence belongs. Therefore, we aim to compress the model from the previous stage to the same size of original models, preserving the model architecture without introducing parameters. By employing a knowledge consolidation approach, we integrated the knowledge from the learning module into the original models. In the consolidation stage, we assume the two separated modules (i.e., original FFN and previous learning module) as a cooperative relationship, where each module is beneficial for translating into different target languages. In the form of distillation, knowledge can be transferred to the distilled model through the distribution (Hinton et al., 2015). And we propose a collaborative distillation to facilitate original components to learn from the learning modules. The collaborative knowledge distillation is given by:

$$\mathcal{L}_{\text{CKD}}(\theta_f, \theta_l) =$$
$$\sum_{y_j, \mathbf{y}_{<j}, \mathbf{x} \in \mathcal{D}',} \text{KL}\Big( P(y_j|\mathbf{y}_{<j}, \mathbf{x}; \theta_f)$$
$$||P(y_j|\mathbf{y}_{<j}, \mathbf{x}; \theta_l)\Big) \quad (6)$$

where $\theta_f$ and $\theta_l$ represent the parameters of original FFN and learning modules, respectively.

In addition to retrospecting the original knowledge, we also intervene in the stage of consolidation to make it relatively smooth. Given the model parameters $\theta$ and the model distribution $p(x|\theta)$, it is natural to maximize the likelihood function to optimize $\theta$. To evaluate the optimization for $\theta$, we define a score function $s(\theta) = \nabla_\theta \log p(x|\theta)$ and its measure of uncertainty that is regarded as the covariance of the models, representing the degree of correlation between two arbitrary variables that change together:

$$\mathbf{F} = \mathbb{E}_{p(x|\theta)}[\nabla_\theta \log p(x|\theta)\nabla_\theta \log p(x|\theta)^\top] \quad (7)$$

In this task, we can approximate the expectation in $\mathbf{F}$ using empirical distribution $\hat{q}(x)$, which is given by the parallel training data:

$$\begin{aligned}\hat{\mathbf{F}} &= \mathbb{E}_{\hat{q}_{\mathbf{x},\mathbf{y}}}[\nabla_\theta \log p(\mathbf{x},\mathbf{y};\theta)\nabla_\theta \log p(\mathbf{x},\mathbf{y};\theta)^\top]\\ &= \frac{1}{N}\sum_{n=1}^{N}\nabla \log p(\mathbf{y}|\mathbf{x};\theta)\nabla \log p(\mathbf{y}|\mathbf{x};\theta)^\top\end{aligned}$$
$$(8)$$

The role of $\mathbf{F}$ is a measure of curvature of the optimization and has a connection to our $\mathcal{L}_{\mathrm{CKD}}$. This gives rise to natural gradient loss $\mathcal{L}_{\mathrm{CKD-F}}$ with the information matrix $\hat{\mathbf{F}}$ which can define the local curvature in distribution space:

$$\begin{aligned}\mathcal{L}_{\mathrm{CKD-F}}(\theta_f,\theta_l) =&\\ \mathcal{L}_{\mathrm{CKD}}(\theta_f,\theta_l) &- \frac{\lambda}{N}\sum_{i=1}^{N}(\theta_f - \theta_f^*)^2 \quad (9)\end{aligned}$$

where $\lambda$ is a hyper-parameter to balance the original parameters and learning modules. $\theta_f^*$ represents the original parameters of FFN with computing the information matrix. Thus we can optimize the original FFN in a smooth region, preserving the previously acquired knowledge. Note that we utilize a small-scale set corresponding to the previous task to calculate the $\mathbf{F}$ matrix.

# 4 Experiments

## 4.1 Experiment Settings

**Parallel Data** In this work, we focus on continual adaptation for MNMT models, aiming to enable the model to improve performance of supervised and zero-shot translation using only new data. To ensure the reliability and reproducibility of the experiments, we provide the German-Chinese (*de-zh*) bilingual data considered for continual adaptation, as the newly available training corpus[2]. Considering the issue of data quality, we performed data cleaning and sampling, resulting in a corpus of 10 million sentence pairs. As a result, we denote many-to-English (*xx→en*) as original translation directions, German-to-Chinese (*de→zh*) and Chinese-to-German (*zh→de*) as new supervised translation directions, many-to-German (*xx→de*) and many-to-Chinese (*xx→zh*) as new zero-shot translation directions. The validation data is from FLoRes (Goyal et al., 2022) and we only choose the validation set in the new supervised translation direction for model selection.

**Model Configuration** In our scenarios, we have chosen to employ the mBART50-nn (Tang et al., 2020) as the base MNMT model. The mBART50-nn is an English-centric multilingual model capable of handling 50 different languages. To ensure consistency, we tokenize the parallel data using the same SentencePiece (Kudo and Richardson, 2018) model as mBART50-nn, which boasts a shared vocabulary consisting of 250,000 tokens. The mBART50-nn is structured as a Transformer-based model, featuring 12 encoder layers, 12 decoder layers, and 16 multi-attention heads. For a more comprehensive understanding of model configuration, additional details are available in Appendix A.

**Baselines** We compare our proposed LCCA with various representative methods in continual learning and transfer learning for continual adaptation. The baselines can be listed as follows:

- Scratch (Vaswani et al., 2017): training a bilingual Transformer-big model from scratch in the supervised translation directions.

- mBART50-nn (Tang et al., 2020): implementing all translation directions based on this original MNMT model directly.

- Fine-Tuning (Luong and Manning, 2015): tuning based on the original models with new data. All original parameters are trainable.

- Mixed-FT (Sun et al., 2019): mixing the external training data related to original translation directions with the new training data to train the model jointly.

---

[2]https://data.statmt.org/cc-matrix/

| Method | Old Directions | New Directions | | AVG. | Old Directions | New Directions | | AVG. |
|---|---|---|---|---|---|---|---|---|
| | $xx{\to}en$ | $de{\to}zh$ | $xx{\to}zh$ | | $xx{\to}en$ | $zh{\to}de$ | $xx{\to}de$ | |
| Scratch | 0.03 | 29.99 | 2.13 | 1.08 | 1.07 | 15.17 | 1.12 | 1.10 |
| mBART50-nn | **24.72** | 3.40 | 5.23 | 14.98 | 24.72 | 6.92 | 8.46 | 16.59 |
| Fine-Tuning | 0.03 | **33.77** | 2.52 | 1.28 | 0.12 | **20.29** | 0.49 | 0.31 |
| Mixed-FT | 19.33 | 29.31 | 6.36 | 12.85 | 20.21 | 15.31 | 4.37 | 12.29 |
| SixT† | 0.21 | 30.33 | 18.48 | 9.35 | 0.98 | 17.22 | 10.92 | 5.95 |
| SixT | 0.14 | 30.05 | 19.40 | 9.77 | 0.72 | 16.85 | 12.04 | 6.38 |
| EWC | 23.01 | 29.93 | 19.47 | 21.24 | 23.19 | 15.54 | 13.97 | 18.58 |
| LFR-CM | 23.11 | 29.91 | 21.13 | 22.12 | 23.13 | 16.74 | 14.97 | 19.05 |
| LFR-OM | 23.03 | 29.97 | 20.93 | 21.98 | 23.32 | 16.13 | 14.74 | 19.03 |
| LCCA w/o LS | 23.47 | 29.98 | 20.30 | 21.89 | 22.97 | 15.23 | 13.74 | 18.36 |
| LCCA | 24.03 | 31.17 | **22.09** | **23.06** | **24.81** | 18.99 | **18.13** | **21.47** |

Table 1: The overall BLEU scores of the particular adaptation directions. "w/o LS" represents that we directly use the consolidation stage without learning stage. "*xx*" denotes the languages supported by mBART50-nn. "AVG" denotes the average BLEU scores on all 100 translation directions. "†" indicates the stage one in the SixT. The highest score is highlighted in **bold** and the second highest score is underlined.

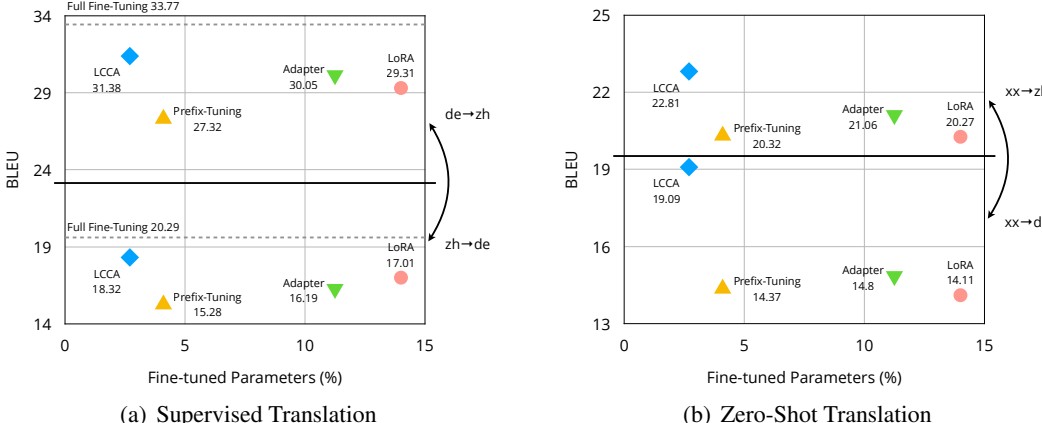

(a) Supervised Translation  (b) Zero-Shot Translation

Figure 3: Performance of parameter-isolation methods for continual adaptation task in multiple translation directions. The x-axis represents the proportion of tuned parameters during fine-tuning relative to the total parameters.

- SixT (Chen et al., 2022): freezing model parameters of encoder and decoder layers in different stages, which is beneficial for zero-shot translations.

- EWC (Kirkpatrick et al., 2017): adding a penalty with Fisher matrix to preserve previous knowledge.

- LFR (Gu et al., 2022): constraining trainable parameters within low forgetting risk regions in continual learning, including LFR-CM and LFR-OM, respectively.

- LoRA (Hu et al., 2022): injecting low-rank matrices into every attention layer, deploying independent tasks of fine-tuned models.

- Prefix-Tuning (Li and Liang, 2021): prepending prefixes to the keys and values at every self-attention layer.

- Adapter (Bapna and Firat, 2019): introducing additional parameters in the FFN layer of original models and freezing all base parameters.

**Training and Evaluation** We evaluate the performance of all translation directions using the FLoRes testsets (Costa-jussà et al., 2022), covering 50 languages from 10 language groups. The performance of translation is measured by the detokenized case-sensitive BLEU score, calculated using the SacreBLEU evaluation script (Post, 2018)[3]. The training time of each method is reported in kiloseconds. All models are implemented using the open-source toolkit fairseq[4] (Ott et al., 2019). For more detailed information, please refer to Appendix B.

[3]Signature: nrefs:1 | case:mixed | eff:no | tok:13a | smooth:exp | version:2.2.0. Many-to-Chinese: nrefs:1 | case:mixed | eff:no | tok:zh | smooth:exp | version:2.2.0.

[4]https://github.com/pytorch/fairseq

| Model | Method | #Size | Translation Directions | | | AVG. | Translation Directions | | | AVG. |
|---|---|---|---|---|---|---|---|---|---|---|
| | | | $xx{\to}en$ | $de{\to}zh$ | $xx{\to}zh$ | | $xx{\to}en$ | $zh{\to}de$ | $xx{\to}de$ | |
| mBART | Base | 0.6B | **24.72** | 3.40 | 5.23 | 14.98 | 24.72 | 6.92 | 8.46 | 16.59 |
| | LCCA | 0.6B | 24.03 | **31.17** | **22.09** | **23.06** | **24.81** | **18.99** | **18.13** | **21.47** |
| M2M | Base | 1.2B | **24.67** | 28.96 | 22.64 | 23.66 | 24.67 | 17.92 | 18.98 | 21.83 |
| | LCCA | 1.2B | 24.00 | **31.88** | **25.42** | **24.71** | **24.82** | **18.84** | **19.76** | **22.29** |
| NLLB | Base | 1.3B | **37.00** | 26.86 | 24.87 | 30.94 | **37.00** | 17.97 | 22.82 | 29.91 |
| | LCCA | 1.3B | 36.13 | **30.99** | **26.37** | **31.30** | 35.98 | **18.97** | **24.27** | **30.13** |
| NLLB | Base | 3.3B | **38.43** | 26.80 | 26.13 | 32.28 | **38.43** | 19.42 | 24.57 | 31.50 |
| | LCCA | 3.3B | 38.17 | **31.35** | **28.93** | **33.55** | 38.02 | **20.03** | **25.43** | **31.73** |

Table 2: The overall BLEU scores based on various MNMT models. "Base" indicates the original performance on MNMT models. The better score of each MNMT model is highlighted in **bold**.

## 4.2 Main Results

As presented in Table 1, we conducted an extensive assessment of translation quality across multiple directions as new data became available. The results indicate that our proposed approach, referred to as LCCA, outperforms several baseline methods, particularly in terms of average BLEU scores for the original many-to-English and new many-to-target translation directions. Specifically, LCCA attains an average BLEU score of 22.09 for xx→zh and 24.03 for xx→en. The trend persists when the target language is switched to German, in which LCCA even demonstrates improvements in the old language translation directions. Although the Fine-Tuning method achieves the best results in supervised translation ($de \leftrightarrow zh$) for continual adaptation, it suffers from catastrophic forgetting, which makes it unsuitable for my scenario. Furthermore, most of the baseline methods are more vulnerable to adapting original models to the new zero-shot translation, while our approach achieves a noticeable improvement. And the results also show that LCCA exhibits no significant difference from the traditional regularization-based approach without the learning phase. Therefore, it involves a process of bringing the added parameters closer to the original parameter space with LCCA.

As shown in Figure 3, we also investigate translation qualities of parameter-isolation methods in multiple directions. The results show that LCCA in the learning stage achieves better performance with fewer additional parameters in both supervised and zero-shot translations for continual adaptation. Moreover, the improvement is more noticeable in the zero-shot translation directions in terms of average BLEU scores, e.g., 22.81 scores on xx→zh and 19.09 scores on xx→de. Although no performance degradation has occurred using parameter-isolation

methods, they lack encouragement to learn new knowledge from updated data, while LCCA in the consolidation stage outperforms the baselines of parameter-isolation, comparing the overall results.

## 4.3 Results on Various MNMT Models

As indicated in Table 2, we have employed a range of pre-trained MNMT models as the initial models and examined the effectiveness of LCCA in the adaptation process. The outcomes underscore the versatility of our approach, showing that it can successfully adapt diverse, powerful models of varying sizes to incorporate newly available data. This adaptability underscores the efficiency and flexibility of our approach in real-world applications. Even if the original models already exhibit strong capabilities in certain translation directions, our method can further enhance the performance through learning knowledge from new data, achieving a comprehensive upgrade in that particular translation direction. We also perform a new comparison between the performance of NLLB 1.3B and 3.3B, and there is no indication that as the model becomes larger, the effectiveness of LCCA reduces significantly.

## 4.4 Ablation Studies

**Effects of Different Learning Modules**

As shown in Table 3, we further investigate different strategies and modules in the learning stage. To learn new knowledge from new data, we introduce the learning modules in the serial or parallel connection manner, representing their relation to the layers of the original models. The results demonstrate that the parallel manner achieves slightly better performance, regardless of whether contrastive learning methods are employed, between 1 and 3 (or 2 and 4). Besides the comparison results between 1 and 2 (or 3 and 4) show that the contrastive

| No. | Learning Module | | New Directions | | | |
|---|---|---|---|---|---|---|
| | Manner | $\mathcal{L}_{\text{CTR}}$ | $de{\rightarrow}zh$ | $xx{\rightarrow}zh$ | $zh{\rightarrow}de$ | $xx{\rightarrow}de$ |
| 1 | serial | ✗ | 31.08 | 22.63 | 18.70 | 18.81 |
| 2 | serial | ✓ | 31.11 | 22.78 | 18.67 | 19.03 |
| 3 | parallel | ✗ | 31.35 | 22.64 | **19.32** | 18.79 |
| 4 | parallel | ✓ | **31.38** | **22.81** | **19.32** | **19.09** |

Table 3: Results on different strategies and modules in the learning stage. The highest score is highlighted in **bold**.

| Method | $xx{\rightarrow}en$ | $de{\rightarrow}zh$ | $xx{\rightarrow}zh$ | $xx{\rightarrow}en$ | $zh{\rightarrow}de$ | $xx{\rightarrow}de$ |
|---|---|---|---|---|---|---|
| Adapter | 24.01 | 29.64 | 20.09 | 23.99 | 16.08 | 13.78 |
| LoRA | 24.00 | 28.79 | 19.45 | 24.26 | 16.70 | 13.12 |
| Prefix | 23.32 | 26.89 | 19.39 | 23.46 | 15.96 | 12.85 |
| Ours | **24.03** | **31.17** | **22.09** | **24.81** | **18.99** | **18.13** |

Table 4: Results on different PET methods in the learning stage. The highest score is highlighted in **bold**.

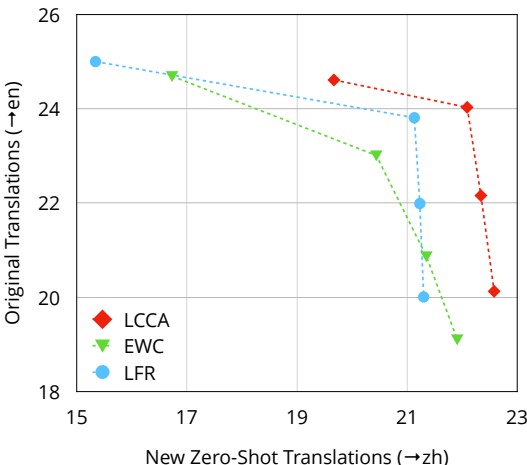

Figure 4: Results using different hyper-parameters. The x-axis and y-axis denote the BLEU of the new zero-shot translations ($xx{\rightarrow}zh$) and original translations ($xx{\rightarrow}en$), respectively. The point closer to the upper right corner indicates better result.

**Effects of Hyper-parameter $\lambda$**

As shown in Figure 4, we compare the consolidation stage with different hyper-parameters $\lambda$ which control the trade-off between the performance of original and new translations. The larger the hyper-parameter $\lambda$, the less the decrease for the previous tasks and the less $\lambda$ is to facilitate to learn new tasks. We compare our method with EWC (Kirkpatrick et al., 2017) and LFR (Gu et al., 2022). For these methods, we search the range from 1 to 0.001. Although these methods are sensitive to hyper-parameters, which restricts the robustness, our method achieves better performance and is relatively less affected by hyper-parameters, according to the downtrend of the curve in Figure 4.

### 4.5 Visualization of Representations

To investigate the zero-shot translations for continual adaptation, we present a visualization of multilingual representations. It studies the shared representations for MNMT models, which can observe their semantic equivalents among multiple languages (Cheng et al., 2022). The results of visualization show that the learning stage of LCCA plays an important role in optimizing multilingual representation spaces and the consolidation stage does not undermine the efforts of the previous stage. The details of the representation analysis are provided in Appendix C.1.

### 4.6 Training Cost

We further investigate the training cost compared with the stronger baselines. The results show that LCCA can improve the efficiency to continually adapt original models to new languages. LCCA can

learning is more important to learn the knowledge from new data for zero-shot translations, and the impact of supervised translation is minimal.

As shown in Table 4, we also supply an experiment to validate the effectiveness of LCCA at the learning phase, compared with different PET-based methods. The results show that the effectiveness of consolidation phase is related to the first learning phase. The higher performance during the learning stage, the better performance on the consolidation stage. Because the role of the consolidation stage is to collaboratively integrate the newly added parameters from the learning stage into the original model.

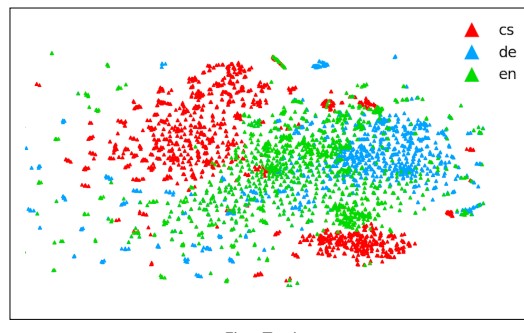

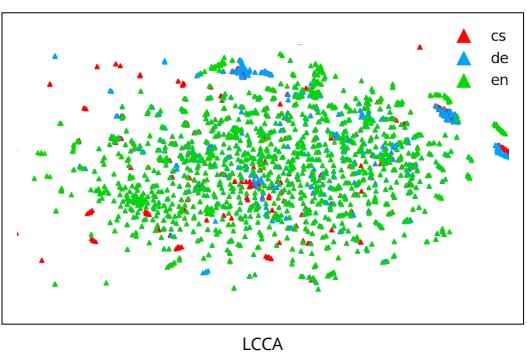

Figure 5: T-SNE visualizations of encoder representations on $xx{\rightarrow}zh$ translations directions.

acquire and consolidate new knowledge efficiently for continual adaptation, which is more practical in the real-world application. Since we only add additional parameters during the two stages in one layer, the training cost does not significantly increase as the depth of the model increases. More details are provided in Appendix C.2.

## 5 Conclusion

In this work, we propose a two-stage method of learning and consolidation to improve performance of pre-trained MNMT models when new data arrives in both supervised and zero-shot translations. It extends multilingual language-agnostic space with the contrastive learning for new data adaptation in the learning stage, and then adopts a collaborative distillation with an information matrix to consolidate both previously and newly learned knowledge for module integration. As a result, our proposed method, LCCA, achieves continual adaptation to multiple translation directions while keeping the model architecture intact. Experimental results demonstrate that the learning stage encourages original models to learn new knowledge from updated parallel data and the consolidation stage mitigates the performance degradation on previous well-performing translation directions when

the model compresses. Further analyses reveal that our method effectively captures linguistic features and bridges the gap of shared representation space for comprehensive continual adaptation.

## Limitations

In this work, we aspire to continually improve performance of MNMT models in translation directions where they are initially weak, and alleviate the issue of degeneration in the well-performing translation directions. In addition to the advantages mentioned, our method does have certain limitations as follows:

(1) Limited available data: We only utilize the parallel data on one language pair for continual adaptation in this work. There are many diverse datasets available, including monolingual data and parallel data on extremely low-resource target language.

(2) Sequential adaptation: This work only considers adapting the original models to new data continually once. However, multiple parallel data are available in a sequential manner. Due to the uncertainty about potential data exposure from the newly available data to the test sets, we plan to carefully design and explore this scenario in future work.

## Ethics Statement

Regarding the datasets, we utilize CC-Matrix as the newly available data and multilingual FLoRes datasets for evaluation (Costa-jussà et al., 2022). All datasets which we use are available in public and widely used by researchers. Similarly, regarding the pre-trained MNMT models, we leverage mBART (Liu et al., 2020), M2M-100 (Fan et al., 2021) and NLLB (Costa-jussà et al., 2022). These models come from the open-source community. The generated process has the potential to show inappropriate and misleading translation results, which can be mitigated by modifying the test sets before inference.

## Acknowledgements

This work is supported by the National Key R&D Program of China (2022ZD0160502) and the National Natural Science Foundation of China (No. 61925601, 62276152). We sincerely thank GT-COM to provide the computing resource and the reviewers for their insightful comments and suggestions to improve the quality of the paper.

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

# A Model Details

We utilize Adam (Kingma and Ba, 2014) $\beta_1 = 0.9$ and $\beta_2 = 0.98$ to optimize the trainable parameters of the MNMT models and adopts the temperature-based sampling (Arivazhagan et al., 2019b) with a temperature of $T = 20$ to balance the training data for the "Mixed-FT" methods. The dropout and attention-dropout are et as 0.3 and 0.1 in all experiments, respectively. For regularization-based method (LFR), we follow the basic setting in Gu et al. (2022). The models are trained on 8 NVIDIA A100 GPUs and we report the mean BLEU scores of the our methods that are trained in three different seeds to ensure results reliable which have randomness. The batch size is 4,096×4 in the procedures of parameter-isolation based methods and 2,048×2 in the procedures of fully tuning methods.

In our experiments, we have implemented several pre-trained MNMT models with varying sizes and configurations. Specifically:

- The mBART50-nn (Liu et al., 2020) model (610 million parameters) comprises 12 stacked encoder layers, 12 stacked decoder layers, and 16 multi-attention heads. The dimensions of hidden states and FFN are 1,024 and 4,096, respectively.

- The M2M-100 (Fan et al., 2021) model (1.2 billion parameters) features 24 stacked encoder layers, 24 stacked decoder layers, and 16 multi-attention heads. The dimensions of hidden states and FFN are 1,024 and 8,192, respectively.

- The NLLB (Costa-jussà et al., 2022) model (3.3 billion parameters) consists of 24 stacked encoder layers, 24 stacked decoder layers, and 16 multi-attention heads, with dimensions of hidden states and FFN set at 2,048 and 8,192, respectively.

For optimizing the trainable parameters of the MNMT models, we employ the Adam optimizer (Kingma and Ba, 2014) with $\beta_1 = 0.9$ and $\beta_2 = 0.98$. We also utilize temperature-based sampling (Arivazhagan et al., 2019b) with a temperature of $T = 20$ to balance the training data for the "Mixed-FT" methods. Dropout and attention dropout rates are both set to 0.3 and 0.1, respectively, in all experiments. During training, the batch size is set to 4,096×4 for parameter-isolation-based methods and 2,048×2 for fully tuning methods with 8 NVIDIA A100 GPUs. And LCCA are trained with three different seeds to ensure the reliability of results considering inherent randomness.

| Code | Language | Family | Script | Order |
|------|----------|--------|--------|-------|
| af | Afrikaans | Germanic | Latin | SVO |
| ar | Arabic | Semitic | Arabic | SOV |
| az | Azerbaijani | Turkic | Latin | SOV |
| bn | Bengali | Indo-Aryan | Bengali | SOV |
| cs | Czech | Balto-Slavic | Latin | SVO |
| de | German | Germanic | Latin | SVO |
| en | English | Germanic | Latin | SVO |
| es | Spanish | Italic | Latin | SVO |
| et | Estonian | Finnic | Latin | SVO |
| fa | Persian | Iranian | Arabic | SOV |
| fi | Finnish | Finnic | Latin | SVO |
| fr | French | Italic | Latin | SVO |
| gl | Galician | Italic | Latin | SVO |
| gu | Gujarati | Indo-Aryan | Gujarati | SOV |
| he | Hebrew | Semitic | Hebrew | SVO |
| hi | Hindi | Indo-Aryan | Devanagari | SOV |
| hr | Croatian | Balto-Slavic | Latin | SVO |
| id | Indonesian | Austronesian | Latin | SVO |
| it | Italian | Italic | Latin | SVO |
| ja | Japanese | Japanesic | Japanese | SOV |
| ka | Georgian | Kartvelian | Georgian | SOV |
| kk | Kazakh | Turkic | Cyrillic | SOV |
| km | Khmer | Austronesian | Khmer | SVO |
| ko | Korean | Korean | Hangul | SOV |
| lt | Lithuanian | Balto-Slavic | Latin | SVO |
| lv | Latvian | Balto-Slavic | Latin | SVO |
| mk | Macedonian | Balto-Slavic | Cyrillic | SVO |
| ml | Malayalam | Dravidian | Malayalam | SOV |
| mn | Mongolian | Mongolic | Cyrillic | SOV |
| mr | Marathi | Indo-Aryan | Devanagari | SOV |
| my | Burmese | Sino-Tibetan | Myanmar | SOV |
| ne | Nepali | Indo-Aryan | Devanagari | SOV |
| nl | Dutch | Germanic | Latin | SVO |
| pl | Polish | Balto-Slavic | Latin | SVO |
| ps | Pashto | Iranian | Arabic | SOV |
| pt | Portuguese | Italic | Latin | SVO |
| ro | Romanian | Italic | Latin | SVO |
| ru | Russian | Balto-Slavic | Cyrillic | SVO |
| si | Sinhala | Indo-Aryan | Sinhala | SOV |
| sl | Slovenian | Balto-Slavic | Latin | SVO |
| sv | Swedish | Germanic | Latin | SVO |
| ta | Tamil | Dravidian | Tamil | SOV |
| th | Thai | Tai-Kadai | Thai | SVO |
| tl | Tagalog | Austronesian | Latin | VSO |
| tr | Turkish | Turkic | Latin | SOV |
| uk | Ukrainian | Balto-Slavic | Cyrillic | SVO |
| ur | Urdu | Indo-Aryan | Arabic | SOV |
| vi | Vietnamese | Austronesian | Latin | SVO |
| xh | Xhosa | Atlantic-Congo | Latin | SVO |
| zh | Chinese | Sino-Tibetan | Han | SVO |

Table 5: The characteristics of languages for evaluation. "S, V, O" represent subject, verb and objective, respectively.

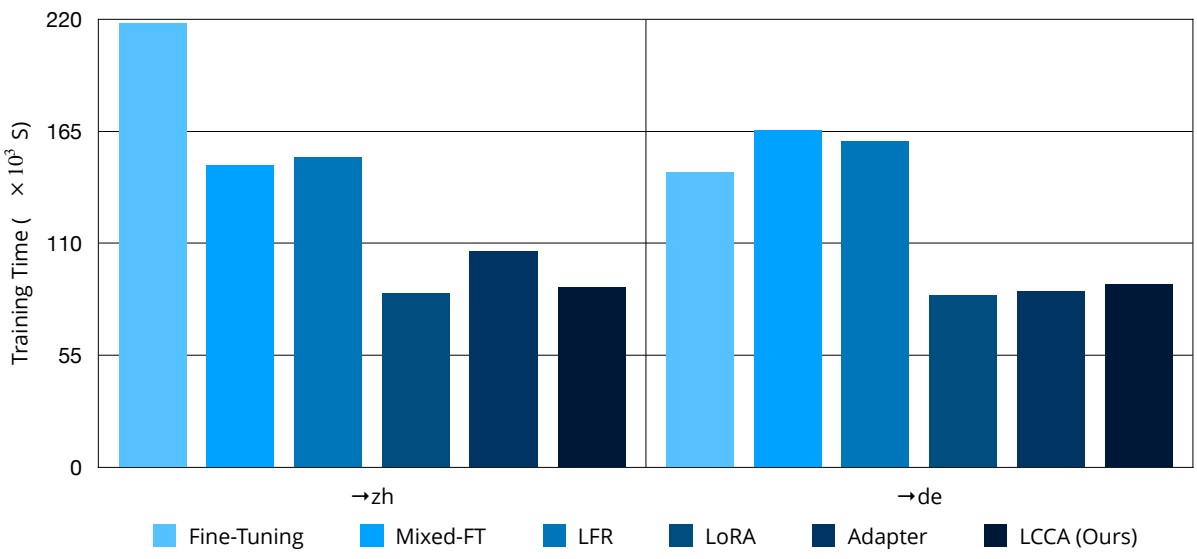

Figure 6: The training time of various methods for adapting mBART50-nn to $xx{\rightarrow}zh$ and $xx{\rightarrow}de$, respectively.

## B Dataset Details

In this work, we utilize parts of the FLoRes de-vtest as our test set, which covers 50 languages and follows the policy in a multi-source setting. The test set contains 1012 sentences in each language and is divided into different language groups with linguistic diversity for evaluation. It is worth noting that a language family refers to a collection of languages that share a common ancestral language, often referred to as the proto-language[5]. Variations in grammar and word order can be observed across different language families[6]. And performing cross-lingual transfer and zero-shot translation between languages with significant differences is challenging. The FLoRes follows the CC-BY-SA 4.0 license that can be freely used for research purposes (Farhad et al., 2021). The characteristics of different languages and language codes used in this paper are shown in Table 5.

## C More Comparisons

### C.1 Visualization of Representations

As depicted in Figure 5, we visualize sentence representations in the context of $xx{\rightarrow}zh$ to probe the representation gap between languages. Achieving comparability within a single representation space necessitates the use of multi-source sentences conveying the same meaning in different languages with t-SNE (Van der Maaten and Hinton, 2008). As evident in Figure 5, the sentence representations produced by LCCA exhibit a closer proximity, illustrating the adeptness of our model at adapting to new data for zero-shot translations. Consistent with this visualization, Table 1 further substantiates that our method excels in achieving improved performance in zero-shot translation scenarios.

### C.2 Training Cost

To further underscore the efficiency of LCCA, we examine the training time in comparison to the more robust baseline models, as depicted in Figure 6. Although our method has two stages, the results show that training time of LCCA is close to that of the parameter-efficient methods and shorter than the other methods with fully tuning, which is more efficient and practical for continual adaptation.

---

[5]https://en.wikipedia.org/wiki/Language_family
[6]https://wals.info