# OpenReview forum: "Learn and Consolidate: Continual Adaptation for Zero-Shot and Multilingual Neural Machine Translation"
_EMNLP/2023/Conference — EMNLP 2023 Main_

### Official Review · Reviewer_ZXCV · 2023-08-05

**Soundness:** 2

**Excitement:**

3: Ambivalent: It has merits (e.g., it reports state-of-the-art results, the idea is nice), but there are key weaknesses (e.g., it describes incremental work), and it can significantly benefit from another round of revision. However, I won't object to accepting it if my co-reviewers champion it.

**Missing References:**

The equation 9 is similar with the equation 5 in [1]. The comparison with this baseline should be added.

[1] Gu et al. 2022. Continual Learning of Neural Machine Translation within Low Forgetting Risk Regions


**Paper Topic And Main Contributions:**

The paper focuses on the problem of continually updating MNMT models with limited new data. They propose a two-stage method consisting of a learning stage and a consolidation stage for continual adaptation (LCCA). The experiments show the effectiveness of the proposed method.

**Questions For The Authors:**

1. It is not clear in Section 3.3. e.g.,

a)	how to get equation 9 from equation 8,

b)	what does $\theta_f^*$ mean in equation 9

c)	In equation 9, what does ‘i’ mean

d)	In equation 9, during inference, which one is used, $\theta_f$ or $\theta_l$

More clarifications and comparisons are needed to evaluate the effectiveness of the proposed method.

2.	The selections of baselines are not reasonable. The key topic in this paper is continuous learning, however, some baselines are unrelated while some baselines are missing. For example, SixT is not designed for the continuous learning. The methods proposed in [1] and [2] should be also considered as baselines.

3.	More experiments should be added in the ablation study. For example, why is FFN selected as the learned module instead of other modules? What are the comparisons results among directly fine-tuning FFN, adapter-based, LoRA-based, prefix tuning method?

4.	Only one language pair de-zh is examined in this paper. More language pairs should be examined.


[1] Overcoming catastrophic forgetting during domain adaptation of neural machine translation
[2] Cross-Attention is All You Need: Adapting Pretrained Transformers for Machine Translation




**Reasons To Accept:**

The research problem and proposed idea are interesting.

**Reasons To Reject:**

The paper needs more clarifications and experiments to better illustrate the details and prove the effectiveness of the proposed method.

**Reproducibility:**

4: Could mostly reproduce the results, but there may be some variation because of sample variance or minor variations in their interpretation of the protocol or method.

**Reviewer Confidence:**

4: Quite sure. I tried to check the important points carefully. It's unlikely, though conceivable, that I missed something that should affect my ratings.

---

> ### Author Rebuttal · Authors · 2023-08-29
>
> Thanks for your review and insights.
>
> > Q1: It is not clear in Equation 9.
>
> We can compute the information matrix to approximate the Hessian matrix with Equation 8 and the information matrix is Integrated into $\theta^{*}_{f}$ in Equation 9.
>
> The $\theta^{*}_{f}$ means the original parameters of FFN with computing the information matrix, and the $i$ means the index of weight matrix. During inference, the $\theta_l$ in the learning phase is integrated into the $\theta_f$, thus we utilize the updated $\theta_f$. We will revise Equation 9 to alleviate any confusion regarding this formula.
>
> > Q2: The selections of baselines are not reasonable. The key topic in this paper is continuous learning, however, some baselines are unrelated while some baselines are missing. For example, SixT is not designed for the continuous learning. The methods proposed in [1] and [2] should be also considered as baselines.
>
> We choose the baselines based on the following reasons.
>
> (1) The key topic in this paper not only the continual learning but also the zero-shot translation (the setting of SixT). Thus, our scenario is an extension of SixT in the context of continual learning. And SixT also mentioned that the approach has the ability to avoid catastrophic forgetting.
>
> (2) The approach of cross-attention adds extra parameters after the encoder, which is not suitable for our scenarios. Further, our consolidation stage can not be directly applied to their method.
>
> (3) The differences between [1] and EWC are not significant, and we believe it won't affect the conclusion of our results. Due to the absence of open-source code and time limitation, we reproduce the method on de-to-zh and will add this baseline in future versions.
>
> | Method | xx->en | de->zh | xx->zh |
> | ------ | ------ | ------ | ------ |
> | EWC    | 23.01  | 29.93  | 19.47  |
> | [1]    | 22.11  | 30.12  | 18.53  |
> | LCCA   | 24.03  | 31.17  | 22.09  |
>
> > Q3: More experiments should be added in the ablation study. For example, why is FFN selected as the learned module instead of other modules? What are the comparisons results among directly fine-tuning FFN, adapter-based, LoRA-based, prefix tuning method?
>
> (1) **The reason for choosing the FFN** is that the FFN might be a core component. The development of Transformer-based language models unveils that the FFN stores the knowledge [2]. External knowledge is injected into FFN layers to enhance the performance of pre-trained language models. Therefore, we consider this aspect as crucial. Thus, we only introduce new parameters in the FFN. Huang et al. [3] also investigate the effectiveness of different modules.
>
> (2) We add an experiment to validate the effectiveness of our method at the learning phase, **compared with different PET-based methods**.
>
> | Method          | xx->en | de->zh | xx->zh | xx->en | zh->de | xx->de |
> | --------------- | ------ | ------ | ------ | ------ | ------ | ------ |
> | Fine-tuning FFN | 12.14  | 31.66  | 3.29   | 0.13   | 19.01  | 0.43   |
> | Adapter-Based   | 24.01  | 29.64  | 20.09  | 23.99  | 16.08  | 13.78  |
> | LoRA-Based      | 24 .00    | 28.79  | 19.45  | 24.26  | 16.70   | 13.12  |
> | Prefix-Based    | 23.32  | 26.89  | 19.39  | 23.46  | 15.96  | 12.85  |
> | LCCA (Ours)     | 24.03  | 31.17  | 22.09  | 24.81  | 18.99  | 18.13  |
>
> The results show that the effectiveness of consolidation phase is related to the first learning phase. The higher performance during the learning stage, the better performance on the consolidation stage. Because the role of the consolidation stage is to collaboratively integrate the newly added parameters from the learning stage into the original model. Moreover, directly fine-tuning FFN may forgets previous knowledge, which makes it unsuitable in our scenario.
>
> > Q4: Only one language pair de-zh is examined in this paper. More language pairs should be examined.
>
> (1) In the test process, we **examine 100 translation directions** for a target language instead of only one language pair.
>
> (2) In the training process, we aim to continually improve performance for both new supervised and zero-shot translation. Therefore, the core idea is how to leverage abundant parallel corpora (X->Y) to enhance the performance of many-to-Y translation. In this paper, **we prodived two target languages with a Latin script and a Chinese script which are totally different**. And in the Limitation section, we have explained the limitations of the current data. We only utilize parallel data on one language pair for continual adaptation in this work. In the future, we will attempt to investigate the effectiveness of our method on extremely low-resource target language.
>
> > Q5: The equation 9 is similar with the equation 5 in [1]. The comparison with this baseline should be added
>
> We have already chosen the [4] as a baseline and make a comparison in Table 1.
>
> [1] Thompson et al. Overcoming catastrophic forgetting during domain adaptation of neural machine translation. In NAACL 2019
>
> [2] Geva et al. Transformer Feed-Forward Layers Build Predictions by Promoting Concepts in the Vocabulary Space. In EMNLP 2022
>
> [3] Huang et al. Knowledge Transfer in Incremental Learning for Multilingual Neural Machine Translation. In ACL 2023
>
> [4] Gu et al. Continual Learning of Neural Machine Translation within Low Forgetting Risk Regions. In EMNLP 2022

---

### Official Review · Reviewer_3Rbs · 2023-08-06

**Soundness:** 2

**Excitement:**

3: Ambivalent: It has merits (e.g., it reports state-of-the-art results, the idea is nice), but there are key weaknesses (e.g., it describes incremental work), and it can significantly benefit from another round of revision. However, I won't object to accepting it if my co-reviewers champion it.

**Paper Topic And Main Contributions:**

This paper focuses on continual learning for multilingual neural machine translation (MNMT). To leverage new knowledge in new available parallel data as well as avoid catastrophic forgetting, the authors propose a two-stage approach: 1) encoding knowledge into a new module and 2) compressing and shifting the learned knowledge into the original model. Results with several open-source MNMT models show improved target-related supervised and zero-shot performance over a couple of strong baselines.


**Reasons To Accept:**

The authors introduce a new approach that facilitates continual learning for MNMT. Improvements were reported in both target-related supervised directions and zero-shot directions.


**Reasons To Reject:**

1) Some statements in this paper are over-claimed.
  * In line 105, "... without introducing additional parameters." However, at the learning stage, a new feed-forward module is required to capture the new knowledge.
  * In line 110, "which improves performance for zero-shot translations in continual learning." This claim is too general: it implies that the proposed approach improves zero-shot translation. But the performance is mainly measured on the translation directions related to the language pairs of the new training data.
2）The proposed approach is not well ablated. Current results don't demonstrate the necessity of the two-stage strategy.
  * In Section 3.2, the authors argue that L_CTR could induce language-agnostic representations. But, in Table 3, we find that L_CTR benefits translation very little. Do we need it?
  * The authors didn't explore the necessity of the separation of the two stages. What if removing the new learnable module and directly optimizing the FFN layer with L_CE and L_CKD-F?
3) Some experimental results are hard to understand.
  * In Figure 3, LCCA outperforms almost all efficient tuning methods in all conditions at the learning stage. But the learning stage only copies the last-layer FFN module. Why would it surpass Adapter and LoRA, particularly considering the latter two have more parameters? Is it because the initialization?
  * In Table 2, as model becomes larger, the effectiness of LCCA reduces. It seems to suggest that model scaling might have a big role here and even change the conclusion.
  * Without distillation, the models from the learning stage already achieves super great performance as shown in Table 3.
4) This paper would benefit from a proof-reading.

**Reproducibility:**

3: Could reproduce the results with some difficulty. The settings of parameters are underspecified or subjectively determined; the training/evaluation data are not widely available.

**Reviewer Confidence:**

4: Quite sure. I tried to check the important points carefully. It's unlikely, though conceivable, that I missed something that should affect my ratings.

---

> ### Author Rebuttal · Authors · 2023-08-29
>
> Thanks for your insightful feedback and constructive comments.
>
> - For each of your concerns on over-claim:
>
> > Q1: In line 105, "... without introducing additional parameters." However, at the learning stage, a new feed-forward module is required to capture the new knowledge.
>
> Thanks for pointing this out. Our goal is to **preserve the original model's architecture** **and keep the number of parameters** after the entire two-stage training process rather than indicating that new parameters are not introduced during the intermediate process. Although our method increases the number of parameters at the first stage, LCCA specifically designs to reduce this added parameters at the second stage. The advantages are: (1) no increase overhead in inference; (2) parameters are non-isolated, eliminating the need to select specific adapters for specific languages. We will modify this misunderstanding in future versions to make it more rigorous.
>
> > Q2: In line 110, "which improves performance for zero-shot translations in continual learning." This claim is too general: it implies that the proposed approach improves zero-shot translation. But the performance is mainly measured on the translation directions related to the language pairs of the new training data.
>
> In this paper, we investigate a particular zero-shot translation proposed by [1], where not using direct parallel sentence pairs are considered as zero-shot translation. In this scenario, the model is trained with parallel dataset of only one language pair (such as de-zh). The trained model is tested on many-to-one language pairs (like cs/fi/fr-zh) in a zero-shot manner.
>
> > Q3: The proposed approach is not well ablated. Current results don't demonstrate the necessity of the two-stage strategy.
>
> Thank you for your constructive suggestion. Just as the title suggests, the second stage serves as a consolidation phase. In principle, it involves a process of bringing the added parameters closer to the original parameter space. If we solely adopt the strategy of the second stage, the model will be optimized from a randomly initialized space. To validate this statement, we conduct the experiment using only the second-stage approach
>
> | Method                  | xx->en | de->zh | xx->zh | xx->en | zh->de | xx->de |
> | ----------------------- | ------ | ------ | ------ | ------ | ------ | ------ |
> | EWC                     | 23.01  | 29.93  | 19.47  | 23.19  | 15.54  | 13.97  |
> | LCCA w/o learning phase | 23.47  | 29.98  | 20.30   | 22.97  | 15.23  | 13.74  |
> | LCCA                    | 24.03  | 31.17  | 22.09  | 24.81  | 18.99  | 18.13  |
>
> The "LCCA w/o learning phase" means that we only adopt the second stage process. The results show that the method exhibits no significant difference from the traditional regularization-based approach without the learning phase. Therefore, we believe that the two-stage approach is necessary.
>
> > Q4: In Section 3.2, the authors argue that L_CTR could induce language-agnostic representations. But, in Table 3, we find that L_CTR benefits translation very little. Do we need it?
>
> (1) **The effectiveness of "L_CTR" in the learning phase.** While the improvement might seem limited on average, in certain directions, there are notable differences in BLEU scores between the methods w/ and w/o "L_CTR". Besides, we counted the **number** of zero-shot translation directions that achieved performance improvement, as the Table shown:
>
> | Zero-Shot Directions | Increased by less than 1 BLEU | Increased by more than 1 BLEU | Passing significance testing (p<0.05) | No improvement | Zero-shot translation directions |
> | -------------------- | ----------------------------- | ----------------------------- | ------------------------------------- | -------------- | -------------------------------- |
> | xx->zh               | 13                            | 25                            | 23                                    | 11             | 49                               |
> | xx->de               | 13                            | 24                            | 19                                    | 12             | 49                               |
>
> The light increase in the average BLEU is attributed to the remaining 18 directions and the average BLEU downplays the impact of "L_CTR". For example, on xx->zh directions, although we achieve performance improvements of more than 1 BLEU in 31 directions with "L_CTR", the directions with limited or no improvement diluted the overall performance.
>
> We will provide these detailed results on all the translation directions in the revised version.
>
> (2) **The effectiveness of "L_CTR" in the consolidation phase.** We also add a further comparsion to validate the effectiveness of "L_CTR". If we use the learning module w/o "L_CTR" in the consolidation phase, the zero-shot performance decline is greater than the learning module w/ "L_CTR". For xx->zh directions, LCCA w/o "L_CTR" degrade to 21.01 average BLEU (reduced by 1.73 compared to the learning phase) in the consolidation phase, and LCCA w/ "L_CTR" degrade to 22.09 average BLEU (reduced by 0.72 compared to the learning phase) in the consolidation phase. For xx->de directions, we can draw similar comparsion (-2.65 vs -0.96). We will add this ablation study in the revised version.
>
> > Q5: The authors didn't explore the necessity of the separation of the two stages. What if removing the new learnable module and directly optimizing the FFN layer with L_CE and L_CKD-F?
>
> As mentioned above (Q3), we have added these additional experiments to validate the significance of the learning phase. Please refer to that statement.
>
> - For each of your concerns on experimental results:
>
> > Q6: In Figure 3, LCCA outperforms almost all efficient tuning methods in all conditions at the learning stage. But the learning stage only copies the last-layer FFN module. Why would it surpass Adapter and LoRA, particularly considering the latter two have more parameters? Is it because the initialization?
>
> Although the adapter and LoRA have more parameters, the additional parameters in their method are evenly distributed across each layer. And some of these layers play a limited role. As you rightly pointed out, the advantage of copying is evident in initialization, which has been demonstrated by [2], compared with the other PET methods. Although it might be better to extend to all layers rather than just the penultimate layer, we propose LCCA to ensure method generality and efficiency for various pre-trained models with differnet sizes.
>
> > Q7: In Table 2, as model becomes larger, the effectiness of LCCA reduces. It seems to suggest that model scaling might have a big role here and even change the conclusion.
>
> Thanks for your insights. The primary intention of this table is to demonstrate that our method can adapt various powerful models of different sizes to newly available data, offering efficiency and flexibility in the real-world. And we will answer your questions from the following three points:
>
> (1) These three models cannot be directly compared since the training data and details for the base models (mBART / M2M / NLLB) differ. To address this, we performed a new comparison between the performance of nllb 1.2B and 3.3B, and there is no indication that as the model becomes larger, the effectiveness of LCCA reduces significantly. For instance, when de->zh arrives, NLLB-1.2B improves 0.31 average BLEU and NLLB-3.3B improves 1.27 average BLEU with LCCA. When zh->de arrives, NLLB-1.2B improves 0.21 average BLEU and NLLB-3.3B improves 0.23 average BLEU with LCCA
>
> | Model | Method | Size | xx->en | de->zh | xx->zh | xx->en | zh->de | xx->de |
> | ----- | ------ | ---- | ------ | ------ | ------ | ------ | ------ | ------ |
> | M2M   | Base   | 1.2B | 24.67  | 28.96  | 22.64  | 24.67  | 17.92  | 18.98  |
> | M2M   | LCCA   | 1.2B | 24.00  | 31.88  | 25.42  | 24.82  | 18.84  | 19.76  |
> | NLLB  | Base   | 1.3B | 37.00  | 26.86  | 24.87  | 37.00  | 17.97  | 22.82  |
> | NLLB  | LCCA   | 1.3B | 36.13  | 30.99  | 26.37  | 35.98  | 18.97  | 24.27  |
> | NLLB  | Base   | 3.3B | 38.43  | 26.80  | 26.13  | 38.43  | 19.42  | 24.57  |
> | NLLB  | LCCA   | 3.3B | 38.17  | 31.35  | 28.93  | 38.02  | 20.03  | 25.43  |
>
> (2) When de->zh arrives, the reduction in the effects you mentioned may not exist between M2M (1.05 improvement) and NLLB (1.27 improvement). Although the improvement of LCCA based on NLLB-3.3B is limited when zh->de arrives. We still achieved improvements in 41 zero-shot directions. The calculation process is as follows: (33.77 [positive] + (-8.37) [negative] / 49) = 0.86). And the average BLEU downplays the comparison.
>
> (3) The reason of better performance on mBART is that the mBART is a English-centric model, the newly available data plays an important role for non-English-centric directions. But M2M and NLLB are non-English-centric and have already encountered a majority of language pairs, making the impact of the newly available data less significant.
>
> We will carefully revise Table 2 to make the paper clearer.
>
> > Q8: Without distillation, the models from the learning stage already achieves super great performance as shown in Table 3.
>
> Although adding extra parameters can improve performance, as pointed out by [3] and the above mentioned, there are limitations to this approach with **introducing extra parameters**. Moreover, the parameters will continue to increase when new languages continue to arrive. Therefore, we believe that maintaining the architecture of the original model is necessary. This leads us to propose a consolidation phase to preserve architecture. And our approach offers greater flexibility – we can decide whether or not to use the second stage. Each option has its pros and cons, and the choice can be made based on the user's actual needs.
>
> > Q9: This paper would benefit from a proof-reading.
>
> Thank you for your comments. We appreciate your suggestion to have the paper proof-read for improvements. We will certainly take your questions into consideration and work towards enhancing the paper's overall quality.
>
> [1] Chen et al. Towards Making the Most of Cross-Lingual Transfer for Zero-Shot Neural Machine Translation. In ACL 2022
>
> [2] Huang et al. Knowledge Transfer in Incremental Learning for Multilingual Neural Machine Translation. In ACL 2023
>
> [3] Gu et al. Continual Learning of Neural Machine Translation within Low Forgetting Risk Regions. In EMNLP 2022

---

### Official Review · Reviewer_HGki · 2023-08-08

**Soundness:** 4

**Excitement:**

4: Strong: This paper deepens the understanding of some phenomenon or lowers the barriers to an existing research direction.

**Paper Topic And Main Contributions:**

This paper studies how to efficiently scale MNMT model to new data without increasing model size. It proposes a two-stage method with learning and then consolidation.
Results show that their approach beats all earlier approach w/ strong pretrained model and on common baseline (FLoRes eval set).

**Reasons To Accept:**

* This work is one of the first to adapt existing MNMT model to new data without hurting the existing performance or increasing model size.
* Their results look very promising. They conduct experiment on strong public available pretrained MNMT weights and eval set.


**Reasons To Reject:**

* There could be more analysis on how the model weight changed and benchmark the two-stage training efficiency (i.e. efficiently adapting model weights without significant retraining)

**Reproducibility:**

4: Could mostly reproduce the results, but there may be some variation because of sample variance or minor variations in their interpretation of the protocol or method.

**Reviewer Confidence:**

3: Pretty sure, but there's a chance I missed something. Although I have a good feel for this area in general, I did not carefully check the paper's details, e.g., the math, experimental design, or novelty.

---

> ### Author Rebuttal · Authors · 2023-08-29
>
> Thanks for your insightful feedback.
>
> > Q1: There could be more analysis on how the model weight changed and benchmark the two-stage training efficiency (i.e. efficiently adapting model weights without significant retraining)
>
> (1) In this paper, to investigate the effectiveness for continual adaptation, we present a visualization of multilingual representations. And we will add more visualizations at different stages. The results of visualization show that the learning stage of LCCA plays an important role in optimizing multilingual representation spaces and the consolidation stage does not undermine the efforts of the previous stage. In principle, it involves a process of bringing the added parameters closer to the original parameter space.
>
> (2) As larger models are typically more substantial and have varying depths, to ensure method generality and efficiency, we make a sacrifice in the learning stage's performance, **where new parameters are only added in the penultimate layer**. Moreover, the effectiveness and trainable parameter amount of the consolidation phase are influenced by the learning phase. We will add more analysis of the two-stage training efficiency with different model sizes and we hope our study could shed light on developing novel continual learning for future research.

---

### Meta-Review · Area_Chair_s4ME · 2023-09-19

**Recommendation:** 5

**Metareview:**

This paper proposes a new approach of continual learning in the context of multilingual neural machine translation (MNMT). It utilizes newly available parallel data for zero-shot translation without harming the already well-performed directions by a two-stage strategy: 1) learning new knowledge from the new data by introducing a new module, and 2) compressing and transferring the acquired knowledge back into the original model. The outcomes, as demonstrated using various open-source MNMT models, improve supervised and zero-shot translation, surpassing the performance of several strong baselines.

Pros:
- To my knowledge, this represents a novel approach within the research domain of continual learning for MNMT. It offers several advantages, such as enhancing both zero-shot and supervised translations without introducing additional parameters post-training. These improvements are substantiated by strong results. The concept is innovative and intriguing. The experimental design demonstrates thoughtfulness and is supported by reasonable justifications.
- Should the paper incorporate the rebuttals, it would serve as an effective mean to enhance MNMT in light of newly available parallel data.
- The paper conducts a concise yet comprehensive literature review.
- It provides a thorough reporting and comparison of various baselines, resulting in a robust conclusion regarding the proposed method.
- Well-written.

Cons:
- There were some unclear details, but these have been thoroughly addressed in the rebuttal phase.

---

### Decision · Program_Chairs · 2023-10-07

**Decision:**

Accept-Main

**Comment:**

This paper proposes a new approach of continual learning in the context of multilingual neural machine translation (MNMT). It utilizes newly available parallel data for zero-shot translation without harming the already well-performed directions by a two-stage strategy: 1) learning new knowledge from the new data by introducing a new module, and 2) compressing and transferring the acquired knowledge back into the original model. The outcomes, as demonstrated using various open-source MNMT models, improve supervised and zero-shot translation, surpassing the performance of several strong baselines.

Pros:
- To my knowledge, this represents a novel approach within the research domain of continual learning for MNMT. It offers several advantages, such as enhancing both zero-shot and supervised translations without introducing additional parameters post-training. These improvements are substantiated by strong results. The concept is innovative and intriguing. The experimental design demonstrates thoughtfulness and is supported by reasonable justifications.
- Should the paper incorporate the rebuttals, it would serve as an effective mean to enhance MNMT in light of newly available parallel data.
- The paper conducts a concise yet comprehensive literature review.
- It provides a thorough reporting and comparison of various baselines, resulting in a robust conclusion regarding the proposed method.
- Well-written.

Cons:
- There were some unclear details, but these have been thoroughly addressed in the rebuttal phase.